# Efficacy of Telerehabilitation Protocols for Improving Functionality in Post-COVID-19 Patients

**DOI:** 10.3390/life15010044

**Published:** 2025-01-01

**Authors:** Jose Luis Estela-Zape, Valeria Sanclemente-Cardoza, Leidy Tatiana Ordoñez-Mora

**Affiliations:** Physiotherapy Program, Faculty of Health, Universidad Santiago de Cali, Cali 760035, Colombia; valeriasanclemente0@gmail.com (V.S.-C.); leidy.ordonez01@usc.edu.co (L.T.O.-M.)

**Keywords:** telerehabilitation, COVID-19, lung injury, functional status, post-acute COVID-19 syndrome

## Abstract

Background and Aims: Telerehabilitation is essential for the recovery of post-COVID-19 patients, improving exercise tolerance, dyspnea, functional capacity, and daily activity performance. This study aimed to describe telerehabilitation protocols specifically designed for individuals with post-COVID-19 sequelae. Materials and Methods: A systematic review was conducted with registration number CRD42023423678, based on searches developed in the following databases: ScienceDirect, Scopus, Dimensions.ai and PubMed, using keywords such as “telerehabilitation” and “COVID-19”. The final search date was July 2024. The selection of studies involved an initial calibration process, followed by independent filtering by the researchers. The selection criteria were applied prior to critical appraisal, data extraction, and the risk of bias assessment. Results: After reviewing 405 full-text papers, 14 articles were included that focused on telerehabilitation interventions for post-COVID-19 patients. These interventions were designed for remote delivery and included exercise protocols, vital sign monitoring, and virtual supervision by physical therapists. The studies reported improvements in physical function, muscle performance, lung capacity, and psychological outcomes. Significant gains were observed in strength, mobility, and functional capacity, as well as reductions in dyspnea, fatigue, and improvements in quality of life, particularly in social domains. Intervention protocols included aerobic, strength, and respiratory exercises, monitored using tools such as heart rate monitors and smartphones. Conclusions: Telerehabilitation positively impacts lung volumes, pulmonary capacities, dyspnea reduction, functionality, muscle performance, and independence in post-COVID-19 patients.

## 1. Introduction

In 2020, a health alert was issued due to the emergence of SARS-CoV-2, the virus responsible for COVID-19, primarily affecting the respiratory system and ranging from mild infections to severe complications [1,2]. The clinical presentation varies by severity: approximately 40% of cases are mild to moderate, presenting symptoms such as fever, cough, and fatigue, without the need for intensive care. Fifteen percent of patients develop severe forms, characterized by significant respiratory distress, requiring supplemental oxygen or mechanical ventilation to support respiratory function [3,4]. In 5% of cases, the disease progresses to a critical state, associated with complications such as acute respiratory distress syndrome (ARDS), acute respiratory infection (ARI), sepsis, and multi-organ failure, which carries a high risk of mortality and necessitates intensive multidisciplinary management [5,6].

The primary entry of SARS-CoV-2 is mediated by the binding of the SARS-CoV-2 spike protein to the human angiotensin-converting enzyme-2 (ACE2) receptor on the cell surface [7,8]. This entry process involves viral binding, endocytosis, and membrane fusion [9,10]. At this point, SARS-CoV-2 specifically targets the ACE-2 receptor on respiratory cells [11,12], again releasing its single-stranded RNA into the cytoplasm. Rapid replication occurs in the 5′ to 3′ direction, resulting in the production of new virions that infect alveolar cells, particularly type I and II pneumocytes [11,12,13]. This process triggers an inflammatory response and uncontrolled cytokine release, leading to lung tissue damage and gas exchange disturbances. The resulting hypoxemia may progress to acute respiratory failure (ARF) or acute respiratory distress syndrome (ARDS) [14,15], conditions that are often associated with the development of fibrotic lesions. These fibrotic lesions occur when lung tissue undergoes scarring due to chronic inflammation, which reduces lung elasticity and further compromises respiratory function.

Therapeutic management in intensive care units (ICUs) for COVID-19 patients includes the use of supplemental oxygen, high-flow nasal cannula, and, in some cases, noninvasive mechanical ventilation [16,17]. In cases of severe hypoxemia and signs of ventilatory failure, invasive mechanical ventilation (IMV) becomes necessary. However, IMV can lead to complications such as ventilator-induced lung injury or ventilator-associated pneumonia, which have been linked to extended ICU stays, averaging three weeks [18,19]. Prolonged immobilization reduces metabolic demand but also causes significant muscle strength loss, ranging from 6% to 40% [20,21]. This results in generalized and respiratory muscle atrophy, decreased lung volumes and capacities, diminished aerobic capacity, and impaired ability to perform daily activities.

Musculoskeletal impairment, pulmonary dysfunction, and reduced aerobic capacity necessitate physical therapy interventions to restore and improve functional independence [22], positively impacting daily activities in postdischarge COVID-19 patients. Telerehabilitation programs have been developed to support remote intervention models, showing improvements in overall muscle strength, lung function, and quality of life compared to patients receiving no intervention.

The COVID-19 pandemic prompted national and international governments to implement measures such as social distancing and isolation to limit the spread of SARS-CoV-2 [23]. While these measures were essential, they affected healthcare delivery systems. The World Confederation for Physical Therapy (WCPT) and the World Health Organization recommended postponing in-home treatments to reduce viral transmission risk. However, recognizing the importance of rehabilitation in improving clinical outcomes and quality of life, different organizations [24] such as the WCPT, the American Physical Therapy Association [25], the Italian Physiotherapy Association [26], the Australian Physiotherapy Association [27], and the Colombian Association of Physical Therapists [28] adopted strategies to continue remote rehabilitation through information and communication technologies, ensuring enhanced protection for vulnerable populations [29].

Telerehabilitation has proven effective in the recovery of post-COVID-19 patients, particularly in improving respiratory function and functional capacity. Studies have reported significant gains in functional capacity, as measured by the six-minute walk test, and a reduction in symptoms such as dyspnea. One study, using a smartphone app-based program that included aerobic exercises, respiratory training, and muscle-strengthening exercises, showed improved outcomes compared to a control group. Furthermore, respiratory muscle training (RMT) protocols lasting six to twelve weeks have been shown to enhance lung function and reduce dyspnea in post-COVID-19 patients [30].

Telerehabilitation provides an alternative when in-person rehabilitation is not possible, especially for patients with mobility limitations, geographic constraints, or pandemic-related barriers [31]. It allows for continuous monitoring and the adjustment of rehabilitation protocols, supporting a personalized recovery approach [32].

The lack of standardized protocols limits the optimization of telerehabilitation effectiveness. Variations in rehabilitation programs—ranging from basic physical activity interventions to more specialized treatments, such as cognitive rehabilitation for post-COVID-19 brain fog—can lead to inconsistent outcomes [33]. This underscores the need for evidence-based protocols to ensure reliable results. Although the short-term benefits of telerehabilitation are reported, further research is needed to evaluate its long-term effects on quality of life and the prevention of chronic disabilities [34]. This systematic review aims to describe telerehabilitation protocols specifically designed for individuals with post-COVID-19 sequelae, a condition associated with complex functional impairments. The novelty of this review lies in its focus on post-COVID-19 protocols and the evaluation of their impact on functional capacity and quality of life, emphasizing the necessity for standardized approaches in this developing field.

## 2. Materials and Methods

A systematic review was conducted following the guidelines outlined in the PRISMA checklist [35] for systematic reviews and the Cochrane collaboration [36]. A registration was created in PROSPERO with the identification number CRD42023423678.

### 2.1. Search Strategy

#### 2.1.1. Search Source

A search was conducted in the following databases: ScienceDirect, Scopus, Dimensions.ai, and PubMed, with a final search date of July 2024.

#### 2.1.2. Search Terms

To develop the search query, standardized language was used with DeCS/MeSH terms and logical operators such as “OR” and “AND”, which allowed for the formulation of the search equation related to the research problem: ((“rehabilitation” OR “pulmonary rehabilitation”) AND (“COVID-19”) AND (“telemedicine” OR “telerehabilitation” OR “telemedicine” OR “telerehabilitation”)). Appendix A summarizes the search queries for each of the databases.

### 2.2. Study Eligibility Criteria

The research question was established as follows:P: Patients post-COVID-19;I: Telerehabilitation programs;C: Any intervention, conventional management, or no intervention;O: Functionality.

The selection criteria for the search were as follows:-Clinical trials, case series, or original articles.

Exclusion criteria during this review were as follows:-Studies that focused on other chronic lung diseases, not specifically COVID-19.-Articles with missing information, theses, expert communications, or editorials.

### 2.3. Study Selection

The selection of studies began with an initial calibration process, followed by independent filtering by the researchers, which facilitated the identification of relevant documents by title and the removal of duplicate articles. A second filter was then applied to exclude articles whose abstracts did not address the review topic. Subsequently, a third eligibility filter, established by the authors, was applied to exclude articles that did not meet these criteria. Finally, a critical evaluation of all articles was conducted based on their study type (cohort, cross-sectional, longitudinal, case-control) to assess the quality and risk of bias.

### 2.4. Data Extraction

Data from the included studies were independently extracted by the authors, considering the abstracts, methodology, results, and conclusions. This process allowed the construction and detailed analysis of the characteristics of the included studies.

### 2.5. Quality Assessment and Risk of Bias Evaluation

The assessment of the studies’ quality was conducted in a blind and independent manner using both the PEDro [37] and MINOR’S [38] scales. The PEDro scale was used to evaluate the quality of controlled clinical trials, focusing on factors such as participant randomization, allocation concealment, blinding in assessments, and the use of standardized outcome measurement methods. The MINOR’S scale was applied to studies that were not controlled clinical trials, assessing elements such as prospective patient inclusion, sample size calculation, and evaluator blinding.

### 2.6. Data Synthesis

A combination of narrative and statistical methods was used to synthesize the collected data. Flowcharts were employed to visually represent the studies that met the research objective, while tables were used to provide detailed descriptions of virtual-based rehabilitation program designs implemented during the COVID-19 pandemic. The predefined inclusion and exclusion criteria were applied to extract pertinent information on rehabilitation protocols for post-COVID-19 patients. This information included the structure of prescriptions, guidelines, and protocols implemented by healthcare professionals within the framework of telerehabilitation or telemedicine models.

## 3. Results

### 3.1. Studies Included

The systematic search yielded 405 results, from which duplicate studies were removed, leaving 316 records for review. After applying the inclusion and exclusion criteria to the full-text documents, 14 articles were included (Figure 1) that addressed the objective of this research.

### 3.2. Characteristics of the Included Studies

The studies included in this review provided insights into the prescription, guidelines, and established protocols for rehabilitation in post-COVID-19 patients, guided by healthcare professionals using the telerehabilitation or telemedicine model. Table 1 presents the main characteristics of the included studies.

### 3.3. Analysis of the Included Studies

The interventions incorporated multicomponent exercise elements within specific parameters. These included an initial warm-up, aerobic or cardiovascular sessions targeting 60% to 70% of the maximum heart rate, and strength training ranging from bodyweight exercises to external loads between 20% and 60%. The protocols also included stretching exercises [39,40,41,44,45,46,49] and respiratory muscle activation exercises [42]. Some interventions offered personalized training [44], added yoga breathing exercises, or included mindfulness practices [48].

Sessions were monitored using tools such as the Borg scale [40,41,44,45,49], heart rate monitors [45], Fitbit devices [39], pulse oximeters [41,45], threshold pressure devices [42], and smartphones [30,40,47,49]. Physiotherapists [39,40,41,42,43,44,47,48,49,50,51,52], therapists, and rehabilitation physicians [30] supervised the interventions. In some cases, primary care personnel were involved [46], while other studies did not specify the supervising personnel [45].

Intervention protocols typically lasted 30 to 45 min and were conducted over varying durations: 4 weeks [35,44], 6 weeks [30,39,40,47], 7 weeks [46], 8 weeks [41,42,50], 12 weeks [52], 500 sessions [48], or 14 days [49].

#### 3.3.1. Interventions Based on Telerehabilitation Compared to Exercise

Both groups demonstrated clinical improvements in pulmonary function, specifically in vital capacity [41]. Statistically significant improvements were observed in dyspnea, fatigue, and social aspects of quality of life (*p* < 0.05) [41]. One study compared functional training with aerobic exercise via telerehabilitation, showing a reduction in fatigue (−6.7 points; 95% CI = −11.9 to −1.3) and increased in functional capacity (2.6 repetitions in 30STS; 95% CI = 0.3 to 4.9) in favor of the functional exercise group [50], though no significant changes were noted in quality of life [50].

Another study made four comparisons involving respiratory muscle training [42], reporting improvements in both groups that used the training device, with an effect size of 0.90 for quality of life and 0.80 for inspiratory muscle strength. A different study showed significant improvements in strength (*p* < 0.001) and functional capacity (*p* < 0.005) among both hospitalized and non-hospitalized patients participating in the telerehabilitation program.

#### 3.3.2. Telerehabilitation Compared with Exercise-Based Intervention vs. Usual Care

Significant improvements in functional capacity, measured by the 30 second sit to stand test (30SCST), were observed compared to the control group, although the difference was not statistically significant (*p* = 0.06) [39]. Another study reported significant changes compared to the control group (*p* < 0.001) [49]. While one study found no significant differences in dyspnea between the groups [39], another reported a statistically significant improvement (*p* < 0.001) [48,49].

Statistically significant improvements in functional capacity were noted using the 6MWT (*p* < 0.001) [30,49,51]. Quality of life also improved significantly (*p* = 0.045) [30] and (*p* = 0.001) [47,48]. However, two studies found no significant differences [39,52], and no differences were observed between groups in pulmonary function [30,47] or oxygen saturation (*p* < 0.001) [48]. One study, using a predictive model, found improvements in physical function (*p* = 0.005) and a reduction in symptom burden (*p* = 0.002) [51]. Another study assessed walking behavior and exercise self-efficacy, reporting significant improvements in the intervention group (*p* < 0.001) [52].

#### 3.3.3. Studies Without a Control Group

In studies without a control group, statistically significant improvements in functional capacity were observed, as measured by the 60 s sit-up test (60STS) (*p* < 0.001) [40,44] and the SPPB test (*p* = 0.00) [44]. Improvements were also noted in the 6MWT (*p* < 0.001) [46] and heart rate adaptations (*p* = 0.012). Additionally, significant enhancements in quality of life were reported (*p* < 0.001) [40,46].

Regarding the risk of bias, blinding was not performed in the evaluations, as the nature of the intervention made it impractical to blind participants. Due to the direct and interactive components of the intervention, such as active participation in rehabilitation exercises or telemonitoring, it was not feasible to conceal the treatment from the subjects, making blinding impossible in this context (Figure 2 and Figure 3).

## 4. Discussion

Telerehabilitation interventions have demonstrated improvements in the functionality of patients with post-COVID-19 sequelae, enabling remote treatment through digital platforms. These interventions facilitate physical and respiratory rehabilitation while maintaining patient safety. Evidence suggests that telerehabilitation enhances lung capacity, mobility, muscle strength, and emotional well-being, leading to improved quality of life. This underscores the need for protocols specifically tailored to the challenges of post-COVID-19 recovery, emphasizing that standardized, evidence-based protocols are essential for optimizing the effectiveness of telerehabilitation in this population.

Studies involving post-COVID-19 patients have identified various exercises, including breathing techniques, bronchial hygiene, active exercises for all four limbs, medium-to-high-intensity strength training using household objects, balance and proprioception exercises, and aerobic capacity training through in-room walking and repeated sit-to-stand transitions [53,54,55]. These exercises align with established protocols for post-COVID-19 rehabilitation.

The studies reported high patient adherence to treatment, with physiotherapists conducting routine remote follow-ups via phone calls, video calls, Skype, WeChat, and device monitoring. Adherence to telerehabilitation programs was likely supported by observed improvements in physical and cognitive function and increased functional independence [30,39,40,41,42,48,49,50,51,52].

Several methodological limitations were identified, including the lack of standardized protocols, demographic heterogeneity among patients, variations in intervention duration, and the insufficient documentation of post-discharge pulmonary or musculoskeletal sequelae from COVID-19. The influence of pre-existing physical conditions on post-intervention outcomes was not adequately addressed, complicating the objective assessment of therapeutic exercise effectiveness. No hospital readmissions were reported, suggesting that the implemented protocols may have prevented systemic decompensations [47,56].

Authors [57] reported significant improvements in cardiovascular-pulmonary function, daily physical performance, and dyspnea perception. However, it remains unclear whether patients experienced muscle weakness or aerobic capacity impairments that affected basic daily activities, or the extent of systemic COVID-19 sequelae, as many assessments relied on subjective evaluations [50].

Our results are consistent with other studies that have demonstrated positive clinical outcomes comparable to conventional in-person rehabilitation for patients with musculoskeletal, neuromuscular, or cardiovascular conditions [58,59]. Additionally, the feasibility of telerehabilitation for patients with post-COVID-19 syndrome has been established, indicating improvements in physical performance and quality of life, which suggest potential for developing future protocols based on these findings [60].

To address methodological gaps in telerehabilitation interventions for patients with post-COVID-19 sequelae, it is recommended to develop evidence-based standardized protocols that account for patients’ demographic heterogeneity and pre-existing conditions. Standardized outcome measures are essential for comparing and generalizing long-term effects. The accurate documentation of post-discharge pulmonary and musculoskeletal sequelae should be prioritized to improve assessment precision. Long-term follow-ups must include objective measurements of physical and cognitive capacity, minimizing reliance on subjective assessments. Future protocol standardization should consider the range of pre-existing conditions to ensure applicability and effectiveness across diverse patient populations and settings.

This systematic review has limitations, including potential language bias, which may have led to the omission of relevant studies on telerehabilitation protocols for post-COVID-19 patients. Future research should focus on these protocols across diverse populations and settings. While this review provides an overview of telerehabilitation protocols for post-COVID-19 patients, it is essential to pursue further research that is more homogeneous, includes a larger sample size, and standardizes outcome measures.

## 5. Conclusions

Telerehabilitation programs that incorporate muscle training, balance, aerobic capacity, and respiratory techniques have been shown to improve aerobic capacity, dyspnea perception, and functional independence in post-COVID-19 patients, highlighting their potential to optimize clinical outcomes and support functional recovery. Assessing the patient’s initial condition is crucial for designing an appropriate rehabilitation plan and enabling continuous monitoring to adjust treatment as needed. Telerehabilitation provides the advantage of improving outcomes without the need for constant in-person intervention. Ongoing research is required to develop standardized evaluation and follow-up protocols to enhance its effectiveness in patient recovery.

## Figures and Tables

**Figure 1 life-15-00044-f001:**
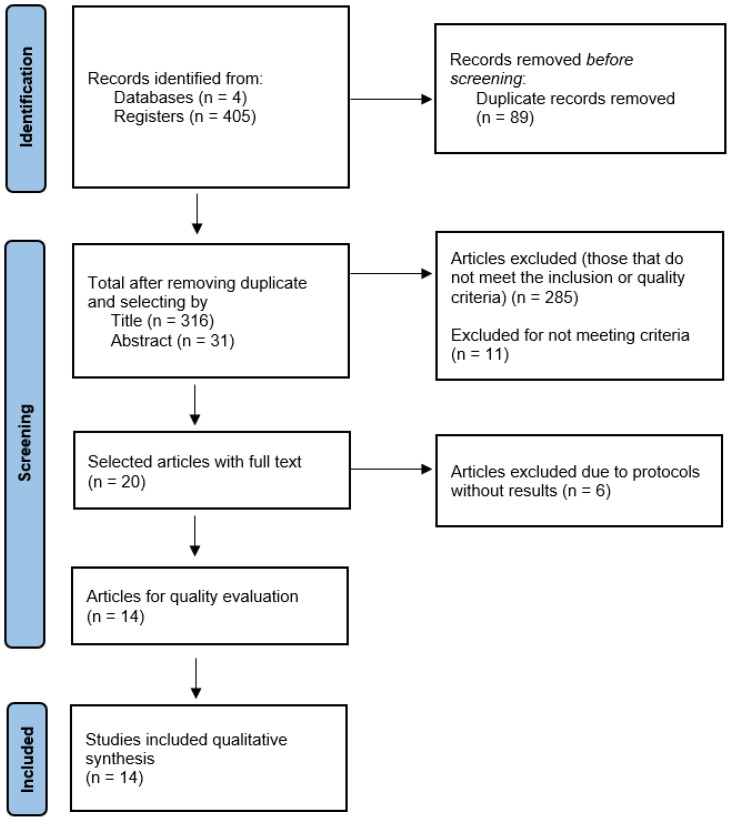
Study selection flowchart.

**Figure 2 life-15-00044-f002:**
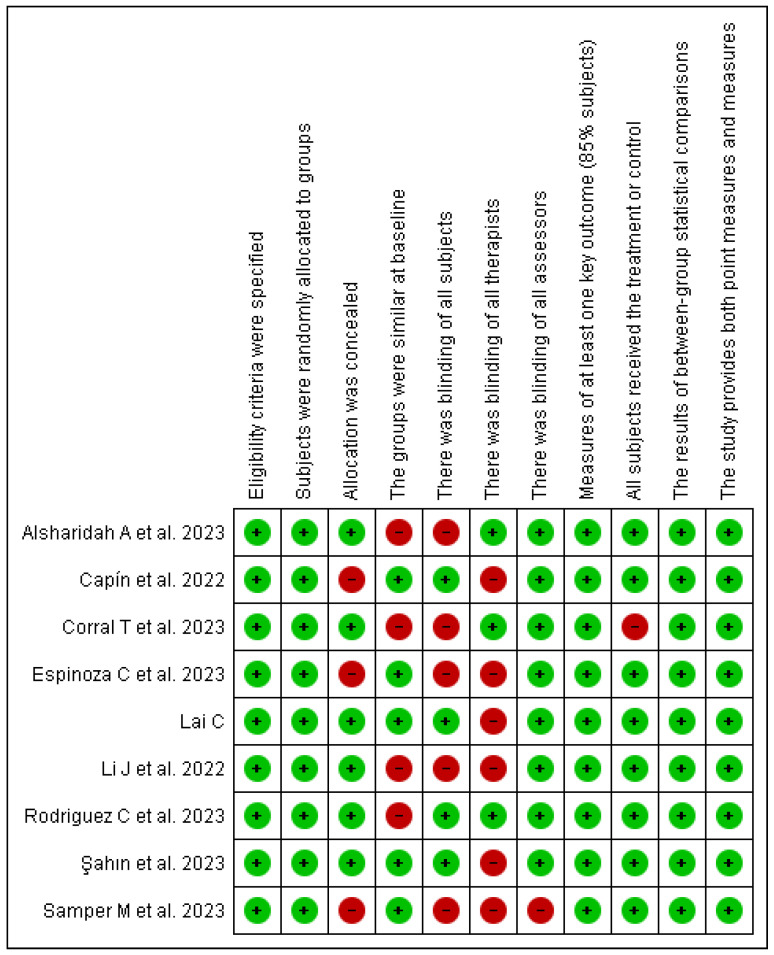
Analysis of the quality of evidence on telerehabilitation in post-COVID-19 patients according to the PEDro scale [39,41,42,43,47,49,50,51,52].

**Figure 3 life-15-00044-f003:**
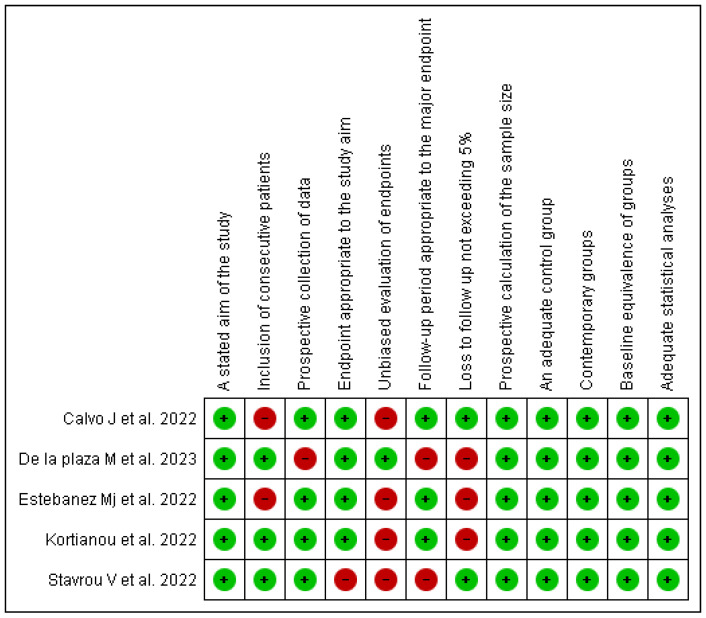
Assessment of the evidence on telerehabilitation in post-COVID-19 patients using the MINOR’S scale [40,44,45,46,48].

**Table 1 life-15-00044-t001:** Characteristics of the included studies.

Authors/Year	Age and Gender	Interventions	Intensity	Scales	Results
Capin et al., 2022 [39]	GI: 29 (54 years)GC:15 (52 years)	GI: Includes breathing techniques (pursed lips, diaphragmatic breathing, yoga), cleansing (snorting), high-intensity strength training (squats, heel lifts, resistance band exercises), and aerobic/cardio activities (walking, cycling, rowing). It also incorporates balance exercises, functional activities like stair climbing, stretching, and lifestyle coaching to promote long-term behavior changeGC: Usual care and education	The exercise program involved 1–5 s per day (5–15 min each), with a maximum of 8 repetitions until failure of strength, low/high intensity based on duration or tempo, and symptom-driven stretching (2–3 sets of 30 s) with a 50–80% success rate goal. All activities were tailored to individual symptoms and goals	30CSTTUGmMRCABC	GI: Mean change = 3.2 (95% CI: 1.8 to 4.6)GC: Mean change = 5.1 (95% CI: 3.2 to 7.0)*p*-value: 0.06 (between groups)GI: Mean change = −1.9 (95% CI: −3.1 to −0.7)GC: Mean change = −0.8 (95% CI: −2.5 to 0.9)*p*-value: 0.21 (between groups)GI: Mean change = −1.5 (95% CI: −2.1 to −0.8)GC: Mean change = −1.4 (95% CI: −2.2 to −0.5)*p*-value: 0.84 (between groups)GI: Mean change = 10.0 (95% CI: 2.7 to 17.3)GC: Mean change = 14.2 (95% CI: 4.5 to 24.0)*p*-value: 0.41 (between groups)


Kortianou et al., 2022 [40]	22 patients (50.1 years)	GI: The warm-up included seated breathing exercises, neck rotations, upper limb movements (abduction, flexion, rotation), and lower limb exercises in supine. Main training focused on strength with progressive weights (0.5–1.5 kg), upper and lower limb exercises, push-ups, and balance training, followed by recovery targeting trunk, upper, and lower limb musclesGC:NR	Exercises started with very light intensity (9–11 RPE) for 8–12 reps over 5–10 min, progressing to 15–20 reps for 15–20 min. Another set involved very light intensity (8–9 RPE) with 2–3 slow reps, holding each for 5–10 min	60STSSF-36	Pre-intervention: Median 22 (IQR: 20–25)Post-intervention: Median 31 (IQR: 25–36)*p* < 0.001PCS: Pre: 37.5 ± 10.3, Post: Mean 52.1 ± 6, *p* < 0.001MCS: Pre: Mean 42.9 ± 11.6, Post: Mean 45.5 ± 12.3, *p* < 0.001
Şahın et al., 2023 [41]	GI: 21 (57.67 years)GC: 21 (63.67 years)	GI: 8-week home-based pulmonary rehabilitation program, including breathing exercises, strength training, and regular walking. The study group received weekly telecoaching calls from a physiotherapist to monitor progress and provide motivationGC: Training only without telerehabilitation	The exercise intensity was guided by a perceived exertion ≤3 on the modified BS	PFT − FEV1 (% predicted)mMRC6MWTSF-36	GI: Pre: 85.05 ± 4.44Post: 89.63 ± 4.96 (0.06)GC: Pre: 83.47 ± 3.92Post: 86.76 ± 4.05 (0.06)GI: Pre: 2.63 ± 0.33Post: 1.21 ± 0.30 (<0.001)GC: Pre: 2.31 ± 0.33Post: 1.84 ± 0.30 (0.62)GI Pre: 378.1 ± 28.9 Post: 440.9 ± 25.8 (<0.001)GC: Pre: 325.1 ± 28.9 Post: 381.7 ± 25.8 (0.001)GI Pre: 52.36 ± 6.25Post: 65.19 ± 4.85Pre: 47.24 ± 5.69Post: 50.91 ± 4.97
Corral T et al., 2023 [42]	G1: 22 (48.9 years) G2: 22 (45.3 years) G3: 22 (46.5 years) G4: 22 (45 years)	G1: IMTG2: IMTshamG3: RMTG4: RMTshamAll interventions used telerehabilitation	G1: 40 min of respiratory muscle training per day using a threshold device for 8 weeksG2: A placebo intervention where participants used a device without resistance to simulate trainingG3: Combined inspiratory and expiratory muscle training with a resistance threshold device, focusing on both breathing phasesG4: A placebo intervention using a device without resistance for both inspiratory and expiratory muscle exercises	MIP MEPIME	G1: Pre: 77.8 (21.6) Post: 109.5 (17.6)G2 Pre: 85.2 (22.8) Post: 98.6 (21.9)G3 Pre: 90.0 (22.1) Post: 123.5 (28.1)G4 Pre: 91.4 (27.2) Post: 104.1 (27.5)G1 Pre: 102.8 (28.6) Post: 123.9 (24.9)G2 Pre: 104.9 (35.6) Post: 116.0 (35.2)G3 Pre: 113.1 (32.0) Post: 154.9 (38.7)G4 Pre: 109.1 (39.9) Post: 127.1 (34.0)G1 Pre: 198.0 (101.1) Post: 456.7 (143.8)G2 Pre: 191.7 (99.3) Post: 322.6 (174.8)G3 Pre: 180.8 (96.9) Post: 459.3 (175.4)G4 Pre: 194.8 (108.5)Post: 308.0 (130.6)
Li J et al., 2022 [43]	GI: 59 (49.17 years)GC: 60 (52.03 years)	GI TERECO intervention groupGC: education	Participants underwent a 6-week unsupervised home-based telerehabilitation program consisting of breathing exercises, aerobic workouts, and lower limb muscle strength exercises, conducted 3–4 times per week via a smartphone app and monitored remotely using heart rate telemetryGC: Participants received brief educational instructions	6 MWDPulmonary Function (FEV1 and FVC)mMRC	GI Pre: 514.52 (82.87), Post difference: 80.20 (74.66), *p* < 0.001GC Pre: 499.98 (93.41), Post difference: 17.09 (63.94)GI Pre: 0.79 (0.14), Post difference: 0.04 (0.17) GC Pre: 0.81 (0.12), Post difference: 0.01 (0.16) *p* = 0.22 GI Pre: 58 (98.3), Post: 90.4 GC, Pre: 58 (96.7), Post: 61.7, *p* = 0.001
Estebanez-Pérez et al., 2022 [44]	GI: 32 (45.93 years) 23 Woman	GI: Exercise with RTGC: NR	GI: 4-week personalized digital physiotherapy program, with 45–50 min daily sessions, 3–5 times per week, consisting of aerobic exercises, strength training, and breathing exercises tailored to their needs	1minSTSSPPB	GI Pre: 14.03 (7.84), Post: 17.53 (7.44), *p* < 0.05GI Pre: 7.90 (1.98), Post: 9.12 (1.69)
Stavrou VT et al., 2022 [45]	GI: 20 (48.9 years)	GI: Exercise with RTGC: NR	GI: 4-week tele-exercise program with 3 sessions per week, each lasting 60 min, including warm-up, aerobic exercise, strength training, and cool-down	6MWTHandgripBorg-CR10	Difference: 32.9 (46.6)Difference: 15.9 (12.3)Difference: 62.9 (42.5)
Calvo J et al., 2022 [46]	GI: 68 (48.5 years)	GI: Exercise with RT	GI: 8-session tele-rehabilitation program over 7 weeks, including patient education, breathing exercises, and physical conditioning with progressive intensity. Exercise intensity was guided by a perceived exertion ≤3 on the modified BS	SF-36mMRC	PF:Pre: 55.6 ± 15.2Post: 11.8 ± 5.3 *p* < 0.001Pre-intervention: 2.57 ± 0.65, Post-intervention: 0.17 ± 0.38
Alsharidah A et al., 2023 [47]	GI: 24 (23.33 years) GC: 24 (22.58 years)	GI: Exercise with RTGC: Usual care and health education	GI: 6-week, home-based pulmonary telerehabilitation program with 60–80 min sessions, three times per week, including aerobics, breathing, and resistance exercises. Exercise intensity was set at 60–80% of maximal heart rate, self-monitored using the Borg scale	6MWTFVCFEV1	GI Pre: 450.83 ± 40.58Post: 514.95 ± 42.96 (0.001)GC Pre: 440.58 ± 51.47Post: 459.12 ± 53.71GI Pre: 2.71 ± 0.41Post: 2.92 ± 0.43GC Pre: 2.72 ± 0.36Post: 2.74 ± 0.37 *p*: 0.12GI Pre: 2.22 ± 0.33Post: 2.39 ± 0.31GC Pre: 2.26 ± 0.26Post: 2.28 ± 0.25*p*: 0.19
De la Plaza et al., 2023 [48]	GI: 50 GC: 50 49 years (IQR = 38–55.75)	GI: Exercise with RTGC: Usual care	22-day respiratory telerehabilitation program with 10 online sessions, including diaphragmatic breathing, chest expansion, and relaxation techniques. Exercises were self-monitored for intensity using a pulse oximeter and the modified BS to ensure safety	Mahler’s Dyspnea IndexQuality-of-life EuroQoL-5D	GI Pre: Median 7 (IQR 5–9), Post: Median 10 (IQR 8–11), *p*: <0.001 GC Pre: Median 6 (IQR 4.75–8.25)Post: Median 7 (IQR 5–10), *p*: 0.001GI Pre: Median 0.74 (IQR 0.52–0.80)Post: Median 1 (IQR 0.79–1), *p*: <0.001CG Pre: Median 0.63 (IQR 0.531–0.739)Post: Median 0.647 (IQR 0.518–0.792), *p*: 0.043
Rodriguez-Blanco et al., 2023 [49]	GI: 24 (38.75 years)GC:24 (42.58 years)	GI: Exercise with TRGC: Daily living activities	The intervention involved a 14-day telerehabilitation program with daily 30 min strength and breathing exercises, performed at home with 12 repetitions per exercise, adjusting intensity based on perceived exertion BS	BSMD126MWT30 STST	GI Pre: 4.87 ± 2.11Post: 0.62 ± 0.65, Difference: −4.25GC Pre: 4.67 ± 1.95Post: 4.42 ± 1.84Difference: −0.25GI Pre: 11.29 ± 7.54Post: 1.08 ± 1.53, Difference: −10.21GC Pre: 10.29 ± 6.82Post: 9.92 ± 6.59, Difference: −0.37GI Pre: 429.63 ± 192.50, Post: 577.54 ± 153.04 Difference: 147.92 GC Pre: 379.46 ± 131.28, Post: 379.08 ± 131.37 Difference: −0.37 GI Pre: 11.63 ± 2.39 Post: 14.71 ± 4.24Difference: 3.08 CG Pre: 10.42 ± 2.48 Post: 10.63 ± 2.70, Difference: 0.21
Espinoza-Bravo et al., 2023 [50]	Experimental group (Functional Exercise—FE): 21 participantsMedian age: 40.9 years (SD: 7.1)Control group (Aerobic Exercise—AE): 22 participantsMedian age: 43.8 years	GI: Functional exerciseGC: Aerobic exerciseBoth groups used telerehabilitation	GI: 8-week, low-intensity functional exercise program (body weight and resistance training) combined with breathing techniques, targeting an exertion level of 4 on the BSGC: low-intensity walking protocol, also combined with breathing techniques, adjusted weekly	SF-366MWTmMRC	PF:Pre: 60.1 ± 15.2 (Experimental), 58.9 ± 14.8 (Control)Post: 75.6 ± 11.3 (Experimental), 61.2 ± 13.1 (Control)Mental Health:Pre: 52.2 ± 10.1 (Experimental), 54.8 ± 9.8 (Control)Post: 65.3 ± 8.7 (Experimental), 55.1 ± 9.3 (Control)Pre: 429.63 ± 192.50 (Experimental), 379.46 ± 131.28 (Control)Post: 577.54 ± 153.04 (Experimental), 379.08 ± 131.37 (Control)Pre: 3.42 ± 2.57 (Experimental), 4.67 ± 2.26 (Control)Post: 1.42 ± 1.84 (Experimental), 4.50 ± 2.15 (Control)
Samper et al., 2023 [51]	GI: 52 (48 years)GC: 48 (48 years)	GI telerehabilitation program with ReCOVery APP.CG: Standard treatment	GI participated in a telerehabilitation program using the ReCOVery APP, which included moderate-intensity physical exercises, respiratory physiotherapy, and cognitive stimulation exercises tailored for long COVID recovery over 12 weeksCG: Standard treatment as prescribed by their general practitioners without any additional rehabilitation interventions	SF-36Sit-to-Stand Test	GI Pre (T0): 29.06 ± 13.673 Months (T1): 33.80 ± 12.19Change (T1–T0): 4.56 ± 12.14GC Pre (T0): 35.58 ± 18.863 Months (T1): 42.30 ± 20.31Change (T1–T0): 8.02 ± 14.38 *p* = 0.021 SF-36 Mental Health:GI Pre: (T0): 32.64 ± 17.983 Months (T1): 37.35 ± 20.01Change (T1–T0): 5.07 ± 16.10GC (T0): 37.09 ± 20.593 Months (T1): 40.29 ± 19.59Change (T1–T0): 3.20 ± 18.27Significance: *p* = 0.491 (no significant difference)GI Pre 9.87 (3.77) Post 10.65 (3.66)GC Pre: 10.92 (3.10) Post 11.28 (3.89)
Lai CY et al., 2024 [52]	GI: 91 (38.9 years) GC: 91 (40.8 years)	GI: telerehabilitation program with moderate-to-vigorous intensity exercisesGC: Usual lifestyle	GI: 12-week TR with moderate-to-vigorous intensity exercises (60–89% heart rate reserve), including aerobic training, tailored to individual fitness levels and monitored via wearable technologyGC: Physical activity counseling and maintained their usual lifestyle without structured exercise interventions	Cardiorespiratory Fitness RFC-VO2peak (mL/kg/min)PAExercise Self-Efficacy (Scale: 1–25)WHOQOL-BREF	GI Pre: 25.8 (6.7), Post: 28.4 (7.2)GC Pre: 27.4 (6.6), Post: 26.0 (5.9)Vigorous PA (VPA, MET/week):GI Pre: 792 (1650), Post: 828.1 (1560)GI Pre: 841.3 (1831), Post: 804.3 (1600)Total PA (MET/week):GC Pre: 4379 (3745), GC Post: 3033 (3106)GC Pre: 3704 (4101), Post: 2059 (2520)GI Pre: 10.5 (3.1), Post: 12.1 (3.2)GC Pre: 12.1 (3.2), GC Post: 10.9 (3.1)Health-Related Quality of Physical Domain:GI Pre: 14.0 (2.2), Post: 14.2 (2.2)GC Pre: 14.2 (2.2), Post: 14.0 (2.2)Psychological Domain:GI Pre: 13.4 (2.2), Post: 13.5 (2.3)CG Pre: 13.5 (2.3), Post: 13.4 (2.2)

GI: intervention group; GC: control group; 30CST: 30 s chair stand; TUG: timed up and go; mMRC: dyspnea scale; ABC: activity-specific balance confidence test; RPE: repetitions per minute; 60STS: 60 s sit-to-stand test; SF-36: short-form 36 health survey; PCS: physical component summary; MCS: mental component summary; BS: Borg scale; PFT: pulmonary function test; 6MWT: 6-minute walk test; RPE: rating of perceived exertion; IMT: inspiratory muscle training; IMTsham: inspiratory muscle training sham; RMT: respiratory muscle training; RMTsham: respiratory muscle training sham; MIP: maximal inspiratory pressure; MEP: maximal expiratory pressure; IME: inspiratory muscle endurance; 6MWD: 6-minute walk distance; FEV1: forced expiratory volume in 1 second; FVC: forced vital capacity, RT: respiratory therapy; 1minSTS: 1-minute sit-to-stand test; SPPB: short physical performance battery; PF: physical functioning; EuroQoL-5D: European quality of life-5 dimension questionnaire; MD12: multidimensional dyspnea-12; 30STST: 30-second sit-to-stand test; RFC: Functional Residual Capacity; PA: physical activity; and WHOQOL-BREF: World Health Organization quality of life—BREF.

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
