# Peer review of "Efficacy of Telerehabilitation Protocols for Improving Functionality in Post-COVID-19 Patients"

_life, 2025, doi:10.3390/life15010044_

Round 1

Reviewer 1 Report

Comments and Suggestions for Authors

I have some suggestions for the abstract: Results: Upon reviewing 405 full-text documents, the study included 20 14 articles that described telerehabilitation interventions for post-COVID-19 patients. All programs 21 were designed for remote delivery and included exercise regimens, vital sign monitoring, and vir-22 tual supervision by a physiotherapist. The studies reported improvements in physical function, 23 muscle performance, and psychological outcomes. I suggest to improve the description of data to clarify them and support the conclusions.

I have several comments on the introduction: 1. Introduction 31 In 2020, a health alert was issued in response to SARS-CoV-2, the coronavirus respon-32 sible for COVID-19, primarily affecting the respiratory system (1,2). The disease presents 33 with varying severity: 40% of cases exhibit mild to moderate symptoms, while 15% de-34 velop severe forms requiring supplemental oxygen or mechanical ventilation (3). Approx-35 imately 5% progress to critical illness, with complications such as acute respiratory dis-36 tress syndrome (ARDS), acute respiratory infection (ARI), sepsis, and multiple organ fail-37 ure (4,5). Authors are kindly requested to emphasize the current concepts about these issues in the context of recent knowledge and the available literature. These articles should be quoted in the References list. References 1. Disruption of CCR5 signaling to treat COVID-19-associated cytokine storm: Case series of four critically ill patients treated with leronlimab. J Transl Autoimmun. 2021;4:100083. doi:10.1016/j.jtauto.2021.100083 2. Radiological-pathological signatures of patients with COVID-19-related pneumomediastinum: is there a role for the Sonic hedgehog and Wnt5a pathways?. ERJ Open Res. 2021;7(3):00346-2021. Published 2021 Aug 23. doi:10.1183/23120541.00346-2021. 3. Cytokine Profiles as Potential Prognostic and Therapeutic Markers in SARS-CoV-2-Induced ARDS. J Clin Med. 2022;11(11):2951. Published 2022 May 24. doi:10.3390/jcm11112951.

Regarding the results, I suggest to improve the quality of the “Figure 1. Study selection flowchart”.

Regarding the conclusions, “5. Conclusions 264 Mixed telerehabilitation programs that incorporate muscle strength training, bal-265 ance, proprioception, aerobic capacity, and respiratory techniques have been shown to 266 improve aerobic capacity, dyspnea perception, and functional independence in post-267 COVID-19 patients. These findings highlight the value of telerehabilitation in enhancing 268 clinical conditions and supporting continued recovery toward greater functional inde-269 pendence”. I suggest to underline the novelty of the study and the future prospects.

Author Response

Dear Reviewer,

Thank you for your comments. The suggestions provided have been considered with the aim of improving the manuscript. The corresponding changes can be found highlighted in yellow, along with the page and line numbers for easy reference.

I suggest to improve the description of data to clarify them and support the conclusions

It is corrected and adjusted. Page 1, lines 21-30

Authors are kindly requested to emphasize the current concepts about these issues in the context of recent knowledge and the available literature. These articles should be quoted in the References list. References 1. Disruption of CCR5 signaling to treat COVID-19-associated cytokine storm: Case series of four critically ill patients treated with leronlimab. J Transl Autoimmun. 2021;4:100083. doi:10.1016/j.jtauto.2021.100083 2. Radiological-pathological signatures of patients with COVID-19-related pneumomediastinum: is there a role for the Sonic hedgehog and Wnt5a pathways?. ERJ Open Res. 2021;7(3):00346-2021. Published 2021 Aug 23. doi:10.1183/23120541.00346-2021. 3. Cytokine Profiles as Potential Prognostic and Therapeutic Markers in SARS-CoV-2-Induced ARDS. J Clin Med. 2022;11(11):2951. Published 2022 May 24. doi:10.3390/jcm11112951.

It is corrected and adjusted. Page 16, lines 349-351,359-361 and 373-375

Regarding the results, I suggest to improve the quality of the “Figure 1. Study selection flowchart”.

It is corrected and adjusted. Page 5

I suggest to underline the novelty of the study and the future prospects.

It is corrected and adjusted. Page 15, lines 328-329

Best regards,

Reviewer 2 Report

Comments and Suggestions for Authors

The manuscript is quite well written. The topic is interesting. I have few suggestions:

1- Abstract. Conclusions: The findings suggest that telerehabilitation has a positive impact on the restoration and improvement of lung volumes, pulmonary capacities, reduction of dyspnea, recovery of functionality, muscle performance, and independence in performing activities of daily living in post-COVID-19 patients. Abstract might be beneficial to include a sentence that briefly summarizes the key findings of the study. This can provide readers with a quick overview of the research. 

2- Telerehabilitation is a valuable tool for the recovery of post-COVID-19 patients, par-72 ticularly in addressing respiratory dysfunction and improving functional capacity. Stud-73 ies have shown positive outcomes in post-COVID patients, including improvements in 74 functional capacity, as measured by the six-minute walk test, and a reduction in symp-75 toms such as dyspnea. I suggest to improve this part and add some references to support all the sentences on this important topic.

3- This systematic review 84 aims to describe telerehabilitation intervention protocols and assess their impact on func-85 tionality in post-COVID-19 patients. 86 I suggest to underline the novelty of the study.

4- 2.6. Data Synthesis 134 Narrative and statistical methods were combined to synthesize the gathered data. 135 The studies that addressed the research objective were depicted using flowcharts, and ta-136 bles were used to describe the design of virtual-based rehabilitation programs during the 137 COVID-19 pandemic. I suggest to improve this description to support the conclusions.

5- 4. Discussion 219 220 Telerehabilitation-based interventions have demonstrated improvements in func- 221 tionality for patients with post-COVID-19 sequelae. These strategies facilitate the contin- 222 uation of treatment remotely through digital platforms, enabling physical and respiratory 223 rehabilitation while maintaining patient safety. Evidence indicates that telerehabilitation 224 protocols enhance lung capacity, mobility, muscle strength, and emotional well-being, 225 with a positive impact on quality of life. However, several key issues remain to be ad- 226 dressed. The discussion section needs to be improved. It is necessary to underline the novelty of the study and clarify the observations obtained by the systematic review.

Author Response

Dear Reviewer,

Thank you for your comments. The suggestions provided have been taken into account with the aim of improving the manuscript. The corresponding changes can be found highlighted in yellow, along with the page and line numbers for easy reference.

Abstract might be beneficial to include a sentence that briefly summarizes the key findings of the study. This can provide readers with a quick overview of the research.

It is corrected and adjusted. Page 1, line 21-28

I suggest to improve this part and add some references to support all the sentences on this important topic

It is corrected and adjusted. Page 2, line 85-96 and page 17 line 421-423

I suggest to underline the novelty of the study.

It is corrected and adjusted. Page 3 line 103-108

I suggest to improve this description to support the conclusions.

It is corrected and adjusted. Page 4, line 165-172

Evidence indicates that telerehabilitation 224 protocols enhance lung capacity, mobility, muscle strength, and emotional well-being, 225 with a positive impact on quality of life. However, several key issues remain to be ad- 226 dressed. The discussion section needs to be improved. It is necessary to underline the novelty of the study and clarify the observations obtained by the systematic review.

It is corrected and adjusted. Page 14, line 267-274 and 304-312

Best regards.

Reviewer 3 Report

Comments and Suggestions for Authors

This sentence needs to be clarified, is it about recovery protocols or multi-study telerecovery programs??

,,This study sought to evaluate the efficiency of tele rehabilitation protocols and their impact on functionality in patients with post-COVID-19,,

If they are telerehabilitation protocols, then the search words are missing

,, telerehabilitation protocol,,

I also think that lines 24-27 should be reworded as follows: The findings demonstrate that telerehabilitation has a positive impact on restoring and increasing lung volumes.

In the introduction, I would suggest mentioning fibrotic lesions after severe SARS COV 2 infection, in mechanically ventilated patients (Long-Term Radiological Pulmonary Changes in Mechanically Ventilated Patients with Respiratory Failure due to SARS-CoV-2 Infection)

Please put the same date in the summary and Material and method:

The final search date was March 17 2024 or July (line 7 and 94)??

Please rewrite line 242-244.

The conclusions must be simplified and written more clearly.

Author Response

Dear Reviewer,

Thank you for your comments. The suggestions provided have been taken into account with the aim of improving the manuscript. The corresponding changes can be found highlighted in yellow, along with the page and line numbers for easy reference.

This study sought to evaluate the efficiency of tele rehabilitation protocols and their impact on functionality in patients with post-COVID-19

If they are telerehabilitation protocols, then the search words are missing

,, telerehabilitation protocol,,

By including the word "telerehabilitation protocol" in the initial searches, an insufficient sample was detected in the database records. For this reason, the word was excluded, obtaining a more generalized search, which allowed more records to be detected.

I also think that lines 24-27 should be reworded as follows: The findings demonstrate that telerehabilitation has a positive impact on restoring and increasing lung volumes.

Suggestion accepted and corrected. Page 1; Lines 28-30.

In the introduction, I would suggest mentioning fibrotic lesions after severe SARS COV 2 infection, in mechanically ventilated patients (Long-Term Radiological Pulmonary Changes in Mechanically Ventilated Patients with Respiratory Failure due to SARS-CoV-2 Infection)

Suggestion is accepted, corrected and reference is attached. Page 2; Lines 54-57.

Please put the same date in the summary and Material and method:

The final search date was March 17 2024 or July (line 7 and 94)??

Inconsistency corrected. Page 1 and 3. Lines 18 and 118.

Please rewrite line 242-244.

The conclusions must be simplified and written more clearly.

The conclusion is adjusted. Page 15; Lines 321-329.

Best regards,

Reviewer 4 Report

Comments and Suggestions for Authors

The article examines the effects of telerehabilitation in patients suffering from post-COVID-19 syndrome. Telerehabilitation programs improved lung capacity, mobility, muscle strength, aerobic capacity, and emotional well-being, positively impacting the quality of life. The article highlights the promising potential of telerehabilitation while emphasizing the need for further development of the method.

To establish the credibility and widespread applicability of telerehabilitation, standardizing protocols, objectively measuring outcomes, and resolving accessibility issues are essential. Current research shows promise, but more homogeneous studies with larger sample sizes are necessary to validate its long-term effectiveness.

Simplifying the table is recommended. For instance, using abbreviated terms (e.g., GI = Intervention Group, GC = Control Group) could save space, with a glossary of abbreviations provided at the bottom of the table. The Results column could be standardized and simplified, e.g., 6MWT: GI +50m, GC +20m (p < 0.05). This would reduce redundancy, streamline information, and save space.

The article could include concrete solutions and recommendations to address the current methodological gaps, measure long-term effects, and achieve future standardization. Without these, it remains difficult to objectively evaluate the effectiveness of telerehabilitation.

The article addresses an interesting and important topic; I recommend its publication.

Author Response

Dear Reviewer,

Thank you for your comments. The suggestions provided have been taken into account with the aim of improving the manuscript. The corresponding changes can be found highlighted in yellow, along with the page and line numbers for easy reference.

Simplifying the table is recommended. For instance, using abbreviated terms (e.g., GI = Intervention Group, GC = Control Group) could save space, with a glossary of abbreviations provided at the bottom of the table. The Results column could be standardized and simplified, e.g., 6MWT: GI +50m, GC +20m (p < 0.05). This would reduce redundancy, streamline information, and save space.

Suggestion is accepted and table 1 is summarized. Pages 5-11.

The article could include concrete solutions and recommendations to address the current methodological gaps, measure long-term effects, and achieve future standardization. Without these, it remains difficult to objectively evaluate the effectiveness of telerehabilitation.

Suggestion is accepted and clarification is made about the effectiveness of telerehabilitation. Page 13; Lines 304-313.

Best regards,

Reviewer 5 Report

Comments and Suggestions for Authors

Dear Authors, a very important contribution to the knowledge on COVID-19. I believe the manuscript can proceed forward following appropriate amendments. Thus, please answer or consider the following:

(1) Introduction, line 39: mention about ACE2-independent mechanisms of SARS-CoV-2 entry. Examples of works: doi 10.3390/v14112535 and doi 10.1038/s41556-024-01388-w.

(2) Introduction, line 62: probably there is a double space mark before “SARS”.

(3) Materials and Methods, lines 88-89: if PRISMA was used then I suggest uploading the document with the checklist as supplementary material or at least in the system for the records of the Editorial Office. Currently, I cannot see any file other than the manuscript itself.

(4) Materials and Methods, line 94: here you mentioned that the final search date was July 2024, but in Abstract there is March 2024 (lines 17-18). So, which one is correct?

(5) Materials and Methods, lines 96-101: the search query is okay but the information in the Abstract (line 17) suggested that only "telerehabilitation" and "COVID-19" were used. Maybe add “for instance” or “e.g.” in the Abstract, when giving examples of keywords?

(6) Materials and Methods, line 100: I cannot see any type of appendix in the manuscript file or even in the system. Please make sure it is included in the final version.

(7) Results, line 162: in my opinion, references should be written “(29-31, 34-36, 38)” instead of “(29, 30, 31, 34, 35, 36, 38)”. There are many other examples where they are provided one by one instead of combined. Moreover, the style is mixed because sometimes the combined form is present in the Discussion. Please update your in-text citations, preferably using the template provided by the journal.

(8) Results, line 206: “6MWT” is explained on first use (before line 206 it is mentioned a few times), but abbreviations such as “30SCST” and “60STS” are not explained. Please double-check all abbreviations in the document.

(9) Decide if you want to use the “MINOR’S scale” or “Minors scale” because there are currently two forms in the manuscript (see lines, e.g., 128, 131, and 218).

(10) Discussion, line 246: all citations should be included in one pair of brackets – “(x, y)” and not “(x)(y)”.

Author Response

Dear Reviewer,

Thank you for your comments. The suggestions provided have been taken into account with the aim of improving the manuscript. The corresponding changes can be found highlighted in yellow, along with the page and line numbers for easy reference.

(1) Introduction, line 39: mention about ACE2-independent mechanisms of SARS-CoV-2 entry. Examples of works: doi 10.3390/v14112535 and doi 10.1038/s41556-024-01388-w.

- Suggestion accepted, modified and reference included. Page: 2; lines 47-48

(2) Introduction, line 62: probably there is a double space mark before “SARS”.

- The assessment is accepted and modified. Page: 2; line 75

(3) Materials and Methods, lines 88-89: if PRISMA was used then I suggest uploading the document with the checklist as supplementary material or at least in the system for the records of the Editorial Office. Currently, I cannot see any file other than the manuscript itself.

- It is articulated after the references.

(4) Materials and Methods, line 94: here you mentioned that the final search date was July 2024, but in Abstract there is March 2024 (lines 17-18). So, which one is correct?

- It is verified and corrected. Page 1: line 18 and page 3 line 118

(5) Materials and Methods, lines 96-101: the search query is okay but the information in the Abstract (line 17) suggested that only "telerehabilitation" and "COVID-19" were used. Maybe add “for instance” or “e.g.” in the Abstract, when giving examples of keywords?

- Indication is accepted and adjusted. Page: 1, line 17

(6) Materials and Methods, line 100: I cannot see any type of appendix in the manuscript file or even in the system. Please make sure it is included in the final version.

- Indication is accepted and an appendix is ​​included at the end of the references

(7) Results, line 162: in my opinion, references should be written “(29-31, 34-36, 38)” instead of “(29, 30, 31, 34, 35, 36, 38)”. There are many other examples where they are provided one by one instead of combined. Moreover, the style is mixed because sometimes the combined form is present in the Discussion. Please update your in-text citations, preferably using the template provided by the journal.

- Indication accepted and modified. Page: 11, line 207

(8) Results, line 206: “6MWT” is explained on first use (before line 206 it is mentioned a few times), but abbreviations such as “30SCST” and “60STS” are not explained. Please double-check all abbreviations in the document.

- It fits and provides clarity. Page 12: lines 233 and 249

(9) Decide if you want to use the “MINOR’S scale” or “Minors scale” because there are currently two forms in the manuscript (see lines, e.g., 128, 131, and 218).

- Accepted and modified. Lines: 157, 160 and 264

(10) Discussion, line 246: all citations should be included in one pair of brackets – “(x, y)” and not “(x)(y)”.

- Indication accepted and modified. Line: 292

Best regards,

Reviewer 6 Report

Comments and Suggestions for Authors In the current study, Estela-Zape and colleagues conducted a systematic review to analyze data published up to July 2024 on the rehabilitation of COVID-19 patients using telemedicine-related approaches. The authors systematized the data accumulated to date on this topic, focusing not only on general approaches to rehabilitation, but also providing detailed information on the exercises used, time periods and results obtained. A detailed analysis of the published literature has shown the high effectiveness of telerehabilitation and revealed a number of limitations that do not allow us to fully establish which approaches using tele-devices have been most effective. In my opinion, this systematic review will be of interest to a wide range of Life readers and can be published after correction of a small number of minor comments:   Line 62 - please delete the extra space between of and SARS-CoV-2   Line 177 and throughout the manuscript - please write p (p-values) in italics.   Fig. 2 - please add a year to the Lai C. reference.   Table 1 - please use either min or minutes, but do not mix them (it is better to use min for compactness).   Table 1, p. 8 - please change Si-to-Stand to Sit-to-Stand   References - please correct references according to the author guidelines (what does "[Internet]" mean if published articles (not websites) are used? (e.g. article [3] published in PLoS One?)

Author Response

Thank you for your comments. The suggestions provided have been taken into account with the aim of improving the manuscript. The corresponding changes can be found highlighted in yellow, along with the page and line numbers for easy reference.

Introduction, line 62: probably there is a double space mark before “SARS”.

It is modified. Page 2, Line 75

References - please correct references according to the author guidelines (what does "[Internet]" mean if published articles (not websites) are used? (e.g. article [3] published in PLoS One?)

It is corrected and adjusted. Page 16, reference 3.

Throughout the manuscript - please write p (p-values) in italics

It is corrected and adjusted. Table 1, pages 5-6

Table 1 - please use either min or minutes, but do not mix them (it is better to use min for compactness).

It is attached. Page 5-11

Table 1, p. 8 - please change Si-to-Stand to Sit-to-Stand 

It is modified. Page 10.

Fig. 2 - please add a year to the Lai C

The previous figure has become figure 3, page 13, The year was adjusted to the reference

Round 2

Reviewer 1 Report

Comments and Suggestions for Authors

The authors edited the manuscript and took my suggestions into account. I have no further comments.